# Third-Generation Therapies for the Management of Psychoactive Substance Use in Young People: Scoping Review

**DOI:** 10.3390/bs14121192

**Published:** 2024-12-13

**Authors:** Gabriela Sequeda, Johan E. Acosta-López, Edgar Diaz-Camargo, Eduardo-Andrés Torres-Santos, Valentina López-Ramírez, Diego Rivera-Porras

**Affiliations:** 1Grupo de Investigación en Modelamiento Científico e Innovación Empresarial, Facultad de Ciencias Jurídicas y Sociales, Universidad Simón Bolívar, Cúcuta 540001, Colombia; 2Centro de Investigaciones en Ciencias de la Vida, Facultad de Ciencias Jurídicas y Sociales, Universidad Simón Bolívar, Barranquilla 080005, Colombia; 3Grupo de Investigación en Estudios Fronterizos, Facultad de Ciencias Jurídicas y Sociales, Universidad Simón Bolívar, Cúcuta 540001, Colombia; edgara.diaz@unisimon.edu.co (E.D.-C.); a_lopez11@unisimon.edu.co (V.L.-R.); 4Corporación Universitaria Minuto de Dios-UNIMINUTO, Cúcuta 540001, Colombia; eduardo.torres.s@uniminuto.edu; 5Departamento de Productividad e Innovación, Universidad de la Costa, Barranquilla 080001, Colombia; drivera23@cuc.edu.co

**Keywords:** university youth, psychoactive substance abuse, third-generation therapies, mindfulness, acceptance and commitment therapy, dialectical behavioural therapy

## Abstract

Introduction: Third-generation therapies (TGTs) have been shown to be effective in the treatment of substance use behaviors in college-aged adolescents. These therapies are based on acceptance, mindfulness and psychological flexibility, which enable young people to change their Psychoactive Substance Use (PSU)-related behaviours, develop coping skills to manage difficult emotions and thoughts, reduce experiential avoidance and maintain long-term abstinence. Objective: To explore the implementation and potential benefits of third-generation therapies, Acceptance and Commitment Therapy (ACT), Dialectical Behaviour Therapy (DBT) and Mindfulness, for the treatment of PSU in college youth. This review includes articles within a 5-year window. Method: A scoping, observational and retrospective review was conducted using the PRISMA method in SCOPUS, PUBMED and Web of Science. Results: eight studies were found, six based on mindfulness, one on dialectical behaviour therapy and one on acceptance and commitment therapy. The results of the studies are promising and emerging for the intervention of the problem. Conclusion: The interventions used show evidence of reducing PSU and other mental health problems such as anxiety, depression and post-traumatic stress. In addition, they allowed patients to increase their well-being and mindfulness.

## 1. Introduction

Psychoactive substances, whether natural or synthetic, have significant effects on the human nervous system, altering individual emotions, perceptions and behaviour [1]. These effects can be found in a variety of forms, such as alcohol, marijuana, cocaine, amphetamines and opioids, and addiction to them is characterised by the manifestation of a range of cognitive, physiological and behavioural symptoms that create a persistent need for use despite the negative consequences for mental health, general well-being and vital functions [2]. Approximately 275 million people worldwide are affected using these substances, with young adults being the most vulnerable, and for this reason this phenomenon has been considered a public health problem with a negative impact on people’s quality of life. Several studies have shown a link between the habitual use of these substances and an increase in morbidity and mortality [3,4,5], as well as an increased risk of neuropsychiatric disorders [6,7,8], a decrease in school performance [9,10], behavioural problems [11], problems in social relationships and an increase in involvement in crime [12,13].

When analysing problems among young adults from a global perspective, an increase in the prevalence of cannabis and cocaine use is evident. According to the UNODC World Drug Report 2020, the prevalence of cannabis use is 28.5% in Europe, 18.1% in the USA, 11.7% in Oceania, 7.8% in Latin America and 4.8% in Asia. Cocaine use is 1.7% in the USA, 3.8% in Europe, 2.3% in Asia, 2.1% in Oceania and 1.7% in Latin America. Since 2020, the daily use of psychoactive substances has increased worldwide, leading to various health problems, including death. The number of deaths associated with the use of these substances is alarming: 107,622 in the USA, 83,000 in Europe, 50,000 in Asia, 1154 in Oceania and 32,000 in Latin America [14,15,16,17]. Opioid use is the leading cause of fatal overdoses, accounting for up to 70% of deaths from this phenomenon in some countries [17].

This situation is exacerbated in educational settings because of the availability of psychoactive substances and difficulties in family supervision, as well as the ease of drug trafficking [18]. According to epidemiological reports, young people are considered a population at risk for the use of psychoactive substances, which is associated with greater negative consequences such as maintenance of substance dependence [17], neurological [19], cognitive [20], organic [21], emotional [22], psychological [23], family and social dysfunction [24]. These are associated with PSU in the university context, with factors such as stress, anxiety, depression, post-traumatic stress disorder (PTSD), among others [25,26]. Students face increased vulnerability to substance use and abuse to cope with personal difficulties [27]. In addition, the university environment presents academic and social challenges that increase various risk factors and contribute to the establishment of problematic consumption patterns [1]. The combination of these variables leads to psychoactive substances becoming a coping mechanism among university students.

A range of interventions are available to moderate or reduce substance use in young people, including medication-assisted psychiatric treatment to reduce craving and relieve withdrawal symptoms [28,29], cognitive behavioural therapy (CBT) [30], 12-step programs [31], the transtheoretical model [32], therapeutic communities [33] and alternative therapies [34]. However, these treatments have shown limitations in their long-term effectiveness and acceptability to young people [2]. In recent years, third-generation therapies (TGTs), also known as contextual therapies, have emerged, focusing on the integration of behaviour modification and acceptance strategies with mindfulness. This is a novel and promising alternative because it focuses on psychological flexibility, emotional regulation and behavioural change, offering a comprehensive and future-oriented approach [35]. Third-generation interventions are based on transdiagnostic approaches [36] that aim to identify the function that maintains the problematic behaviour, in order to generate greater psychological flexibility, radical acceptance and behavioural change through the modification of verbal contexts by means of metaphors and experiential therapeutic strategies, behavioural modification, behavioural training, among other techniques, with beneficial effects in terms of reducing psychological stress and problematic use of psychoactive substances [1].

The effects of TGT have been compared with other traditional therapies, such as CBT or group therapy, and have shown that TGT is an effective option compared with conventional substance use treatments for young people in both the short and long term, reducing psychoactive substance use, improving quality of life and maintaining improvements during follow-up [1,28,29,30,31,32]. This raises the question: what are the implementation and potential benefits of third-generation therapies for the treatment of psychoactive substance use in young university students?

## 2. Materials and Methods

The PRISMA 2020 methodology, widely recognised in the scientific community as the standard protocol for the development of scoping reviews and meta-analyses, was used to conduct this review. This approach allowed detailed documentation of the selection of each article, including information on authors and results [37]. Studies published in English between 2018 and 2024 were included. The search started in November 2023 and ended in February 2024. The databases selected were PUBMED, SCOPUS and WOS. The information was retrieved using search algorithms constructed with key terms consulted in DECS and MESH. The Population, Intervention and Outcomes (PIO) tool [38] was used to construct the search question (see Table 1).

### 2.1. Electronic Dictionaries, Search Algorithms and Databases

An exhaustive search was conducted on third-generation therapies (mindfulness, ACT and DBT) for the treatment of psychoactive substance use in young university students. Specific terms from the Descriptors in Health Sciences (DECS) and Medical Subject Headings (MESH) thesauri were used (see Table 2). Terms were combined using logical connectors such as AND, OR and NOT, and symbols such as “ ” and () (see Table 2 and Table 3). Search engines such as SCOPUS, PUBMED and Web of Science were used. Filters were used to select articles, clinical trials and randomised trials. The results cover the last 5 years to compile recent and relevant studies on the subject.

### 2.2. Eligibility Criteria

Studies were selected based on inclusion and exclusion criteria that facilitated the assessment of quality and reliability to answer the question posed. The types of articles selected were experimental, quasi-experimental, case study and case control, with psychological intervention based on third-generation therapies. Articles on work with children or adolescents, primary or secondary school students, other types of intervention from other psychological models, assessment of psychoactive substance use, other psychological problems, validation of scales, books or manuals or intervention with another population were excluded.

A total of 128,239 articles were identified and screened; of these, 8 articles met the following criteria:Contained the words of the search equation in the title, abstract, keywords or study variables.Studies of interventions for psychoactive substance use in young university students from Mindfulness-based therapies, Behavioural Activation Therapy or Dialectical Behavioural Therapy.Clinical trials.

Next, the information was filtered and the selected articles were read and analysed by reviewing aspects such as the title, abstract, introduction, methods, results and discussion, which allowed the quality and reliability of the selected studies to be assessed (see Table 4, consolidation table).

The aim of the scoping review is to describe the implementation and benefits of therapies based on mindfulness, behavioural activation therapy and dialectical behaviour therapy in young university students who use psychoactive substances.

The table shows the results of a search for information on the effectiveness of third-generation therapies for the treatment of psychoactive substance use in young university students, carried out in three databases: PUBMED, SCOPUS and Web of Science. A total of 128,239 files corresponding to the search terms were found. In detail by database, 55,442 documents were found in PUBMED, 89 in SCOPUS and 72,708 in Web of Science. The search was restricted to documents from the last five years. In the filtering process, documents were excluded because they were incomplete or duplicated; in PUBMED, one document was discarded. Similarly, documents without access were excluded: 380 in PUBMED, 22 in SCOPUS and 16,001 in Web of Science. Documents that did not match the search criteria were also discarded: 370 from PUBMED, 25 from SCOPUS and 19,826 from Web of Science. After filtering, the final sample of documents was one document from PUBMED, four from SCOPUS and three from Web of Science.

In general, the largest number of documents was found in PUBMED, but it also had the largest number of eliminations due to non-compliance or lack of access. Web of Science had the highest number of documents excluded for non-compliance with variable criteria. SCOPUS had the lowest number of final documents, which may indicate that this database has a low coverage of publications on this topic. A summary of the article selection process is shown in the PRISMA flowchart (Figure 1).

## 3. Results

This section details the studies found on third-generation psychological treatments for the treatment of psychoactive substance use among young people in university contexts. A descriptive analysis of the selected articles is carried out, providing information on, among other things, the therapeutic effect, intervention strategies and duration.

Table 5 shows the characteristics of the studies relating to the use of mindfulness (n = 6), acceptance and commitment therapy (ACT) (n = 1), dialectical behaviour therapy (DBT) (n = 1) therapeutic strategies with young university students who reported having used a psychoactive substance. The methodology used refers to randomised control, quasi-experimental non-randomised, randomised controlled trial and case study designs. The population samples are characterised by being young men and women aged 18 years and over. The results of the studies suggest that Mindfulness, ACT and DBT strategies have enabled the population to increase their well-being and mindfulness, reduce states of anxiety, depression, stress and post-traumatic stress, and decrease the use of psychoactive substances.

### 3.1. Characteristics of the Studies

The most reported therapeutic strategies (n = 6 studies) were mindfulness, with guided meditations [46], loving-kindness meditation [47], navigating with urgency technique, compassionate letter, STOPP technique [48], ultra-brief breath counting intervention [37] and guided mindfulness short meditations [38]. In addition, the use of guided mindfulness meditation apps using smartphones has been described [43]. The results of these interventions suggest reduced stress, anxiety and depression, higher levels of mindfulness, greater self-compassion, resilience, improved sleep quality, greater life satisfaction and general well-being.

From Acceptance and Commitment Therapy, one study was selected, which included sessions of values exploration, cognitive defusion, informal mindfulness exercises, use of metaphors, increased psychological flexibility through experiential activities and intervention with psychoactive substance use. This intervention showed effects on emotional acceptance, psychological flexibility, identification and clarity of personal values, stress coping skills, improved interpersonal relationships, reduced experiential avoidance, commitment to action, decision making, life satisfaction, reduced symptoms of depression and anxiety and satisfaction with treatment [43].

On the other hand, one article was retrieved for analysis from Dialectical Behavioural Therapy, which aimed to strengthen stress coping strategies through training in mindfulness skills, emotional regulation, discomfort tolerance and interpersonal effectiveness, to reduce the stress response and its negative consequences, such as the use of psychoactive substances. Results reported that participants achieved reduced stress levels, increased use of adaptive coping strategies, greater satisfaction, reduced anxiety and depression, improved academic performance and reported maintenance of improvement at 3-month follow-up [41].

The eight studies focused on problematic use of psychoactive substances among young university adults. Psychoactive substance use was addressed together with other difficulties such as stress, post-traumatic stress, anxiety, depression, positive and negative affect, impulsive behaviour, self-improvement and others. In general, the research reported decreases in psychoactive substance use and craving, decreases in negative affect and problematic emotional states such as anxiety or depression, decreases in self-blame and avoidance. Increases in personal well-being, mindfulness, tolerance of discomfort and improved coping strategies were also reported.

In terms of research methods, there is evidence of the use of randomised control designs in three studies [46,47,48], five studies with quasi-experimental designs [38,47,48,49] and one case study [43]. The sample of each study also shows that the minimum number of participants ranged from 1 to 205 young adults in educational settings. To ensure group persistence, two of the studies used measures such as telephone follow-up, delivery of rewards such as chocolate [36] or USD 60 gift cards [45] to those who persisted and at the end of the total number of procedures. Similarly, the number of sessions within the research procedures ranged from 1 session [36], 4 weeks [47], 8 weeks [48,50], 10 weeks, 12 sessions [43], and a weekly meeting during a 13-week academic term [45].

### 3.2. Results of the Effectiveness of the Interventions Used

In terms of research findings, effect sizes are reported ranging from low to high effect [38,46,50] and moderate to high effect [45], low effect [39], significant effect with no effect size reported [44] and no effect size reported because it was a case study [43] (see Table 6). Changes reported in the articles include reduced alcohol or nicotine use, increased alertness, improved well-being, reduced anxiety, depression and stress, improved mood and intention to continue attending therapeutic groups, all due to the interventions (see Table 5).

### 3.3. Limitations of the Studies

Limitations identified in the studies include smaller than expected sample sizes [48] and lack of randomisation in the studies of [38,47,49], which could affect causal inferences. In addition, in one of the studies [47], the intervention was delivered by the same investigator, which may have biased the results. Similarly, Cotter et al. [42] noted difficulties in generalizing the intervention and the need for tighter control of participants’ knowledge of the purpose of the study. Finally, Hogarth et al. [36] highlighted the lower efficacy of breath counting in heavy drinkers and the lack of evaluation of other interventions.

In conclusion, the primary limitations encompass the limited number of sessions, case study designs [43], and the short duration of interventions, all of which hinder variable control and generalization of results. These findings underscore the necessity for further research to comprehensively assess the impact of Mindfulness-based therapies, Acceptance and Commitment Therapy, and Dialectical Behavioural Therapy in populations with psychoactive substance use disorders.

### 3.4. Bias Analysis

The analysis of the reviewed studies reveals several methodological biases that limit the reliability and generalizability of their findings. First, there is evidence of selection bias, which is common in quasi-experimental and single-case studies, such as those by Ehman and Gross [43] and Foley and Lanzillotta-Rangeley [39], where the lack of random assignment limits the ability to infer causality and generalize results to larger populations. This lack of randomisation introduces the possibility that external factors, rather than the interventions themselves, may be responsible for the observed effects.

Additionally, some studies show publication bias by reporting significant results without including effect sizes, as seen in Foley and Lanzillotta-Rangeley [39] and Vinci et al. [44]. This omission of quantitative data makes it difficult to compare results across studies and may create a skewed perception of the effectiveness of the interventions. Attrition bias is also noted in studies such as Rosky et al. [45] and Hogarth et al. [36], where there is insufficient clarity on how missing data were handled, impacting internal validity by not accounting for potential participant dropouts during follow-up.

The combination of these biases, along with small sample sizes in studies like Nguyen et al. [41] and Valenstein-Mah et al. [40] and the lack of extended follow-up in studies such as those by Cotter et al. [42] and Vinci et al. [44], highlights the need for greater methodological rigor in future research to ensure that the results accurately reflect the effectiveness of third-generation therapies in this population.

### 3.5. Analysis of the Articles and the Flaws of the Chosen Articles

Cotter et al. [42]: This study reports an effect size range of 0.2 to 0.9, indicating a low to high effect of Mindfulness intervention on reducing alcohol use among young adults. However, it is important to consider the limitations of the study, such as the small sample size and difficulty in generalizing the results to larger populations.

Ehman and Gross [43]: This study reports significant effects, but it is not possible to generalize the results due to the absence of a control group, because it is a case study and no follow-up assessments are reported, which prevents knowing whether the effects were maintained.

Foley and Lanzillotta-Rangeley [39]: This study does not report the effect size; however, it reports reductions in depression, anxiety and stress. Similarly, it is considered that the intervention was short, since it lasted 10 days and the sample was small (n = 33). Additionally, since it was a mobile application intervention, details about the implementation by the participants are unknown.

Nguyen et al. [41]: This study indicates an effect size greater than 0.2, understood as a low effect. It is reported as quasi-experimental research, i.e., no randomisation of the groups is performed. In addition, the intensive intervention methodology is reported, which means that the process requires accompaniment by the staff and administration of the academic entity.

Rosky et al. [45]: The reported effect size ranges from 0.66 to 1.32, suggesting a moderate to high effect of Mindfulness intervention in improving emotional well-being and reducing stress among young adults. Despite this, the lack of randomisation in the study design and the possible bias of the researcher as a mindfulness instructor may affect the interpretation of the results.

Hogarth et al. [36]: This study presents an effect size range of 0.2 to 1.0, indicating a low to high effect of Mindfulness intervention on stress reduction and alcohol consumption among young adults. However, limitations include the lack of evaluation of other interventions and the need for further research to assess the effect on heavy drinkers.

Valenstein-Mah et al. [40]: An effect size between 0.23 and 0.88 is reported, suggesting a low to high effect of Mindfulness intervention on reducing post-traumatic stress symptoms and alcohol consumption among young adults. However, limitations such as the smaller than expected sample size and limited assessment of alcohol consumption may influence the generalizability of the results.

Vinci et al. [44]: This study has limitations in terms of sample size, quasi-experimental design, non-randomisation of the sample, no follow-up of effects, short duration of the intervention and difficulty in clarity of the variables measured by the instrument.

## 4. Discussion

The aim of this review was to determine the implementation and benefits of third-generation therapies, ACT, DBT and mindfulness, in the treatment of psychoactive substance use in young adults in educational settings. Therefore, the evidence presented in this scoping review suggests that mindfulness-based therapy is a widely used and positive intervention for young people in educational settings. The effect sizes reported range from large to moderate to small, with outcomes including reduced psychoactive substance use, particularly reduced alcohol consumption, increased life satisfaction, reduced anxiety, depression, and stress, improved mood and the intention to continue attending therapeutic groups [46,47,48,49,50]. This confirms that the use of mindfulness can have positive effects on people who use psychoactive substances, improving their functioning, reducing symptoms and contributing to relapse prevention [39,44,45].

Similarly, the research reviewed describes psychological improvements in young people, with some studies showing significant effects with low numbers of sessions, ranging from one intervention session to four sessions, to one session per week during the academic term. Increased psychological flexibility, acceptance of private events, openness to life experiences, improved academic performance, improved coping strategies and enhanced mental health and well-being in individuals reporting psychoactive substance use in an academic context were reported [46,47,48,49,50]. Specifically, in the case of addictions, mindfulness has shown a direct relationship between mindfulness and abstinence [51].

Regarding dialectical behaviour therapy (DBT), it has been shown to be an effective intervention for treating borderline personality disorder, severe disorders and suicidality [47]. Despite this, the results of the present review indicate insufficient evidence for the treatment of psychoactive substance use among young people in academic contexts. This can be attributed to the methodological challenges faced in these studies, such as quasi-experimental designs, lack of randomisation and small sample sizes. However, the study by Nguyen et al. [41] showed significant improvements in stress mindset and coping strategies related to problem-solving skills, social support, humor, spirituality, relief and distraction. They also reported reductions in avoidance, psychoactive substance uses and self-blame, with effects observed even six months after the intervention.

Although the results of the previous study are promising, demonstrating the effectiveness of the DBT strategies used, further research is needed to confirm the therapy’s effectiveness in young university students with psychoactive substance use. As this therapy holds potential benefits for young people, it can contribute to personal and academic development, reduce problematic behaviours, alleviate symptoms, strengthen coping strategies for life challenges and help build a life worth living [51].

In the case of Acceptance and Commitment Therapy (ACT), the evidence for treating substance use in young people in academic contexts is sparse. In the case study by Ehman and Gross [43], significant reductions in alcohol consumption, acceptance of emotions related to sadness and anxiety without resorting to alcohol, reduction in symptoms of depression and anxiety, increased cognitive flexibility, adaptation to life challenges, identification of personal values, improvement in self-esteem and self-confidence, healthy coping with stress, improvement in interpersonal relationships, reduction in experiential avoidance, increase in value-committed behaviour and increase in life satisfaction were reported. Satisfaction with treatment was also noted, with no evidence of a relapse in alcohol use at the six-month follow-up. However, as this research uses a case study methodology, it is not possible to generalize the results, draw conclusions about external validity or compare the treatment’s effectiveness with a control group of young people who did not receive any treatment. Nevertheless, the results of this case study provide a starting point for future research.

In this context, it has been shown that the effects of third-generation therapies demonstrate greater long-term efficacy than immediate post-treatment improvements, indicating that the benefits of these interventions may continue to increase even after the treatment ends [36,40].

Regarding the limitations of the analysed studies, difficulties were observed in the use of objective methodologies, as some studies relied on self-report measures not compared with quantitative measures, potentially leading to symptom overestimation by participants [46,47,48,49,50]. Additionally, in some studies, post-intervention measurements were not taken, hindering knowledge about the maintenance of improvements [46,48,49]. The use of mobile applications [39] and brief interventions [46,47], coupled with small sample sizes, limits the generalizability of results [47,48,49,50]. In general, the studies exhibit various limitations, such as a lack of detailed description of the outcome assessment process, difficulties in controlling variables, unclear criteria for determining the interval between interventions and the presence of a single researcher acting as both evaluator and therapist.

Nevertheless, the reviewed studies reveal methodological limitations that must be considered when interpreting these results. Many studies relied on quasi-experimental designs and case studies without randomised controls, such as those by Ehman and Gross [43] and Foley and Lanzillotta-Rangeley [39], which limits the ability to attribute observed effects solely to the interventions and generalize them to larger populations. Furthermore, some studies presented publication bias, as they reported significant outcomes without specifying effect sizes, as seen in Foley and Lanzillotta-Rangeley [39] and Vinci et al. [44]. The absence of these quantitative measures restricts the ability to compare across studies and could lead to an inflated perception of therapeutic effectiveness. Additionally, attrition bias was noted in studies where handling of missing data was not clearly explained, such as in Rosky et al. [45] and Hogarth et al. [36], which impacts internal validity by not accounting for participant dropout during follow-up periods.

Another limitation relates to sample sizes and intervention durations. Many studies utilized small samples, limiting statistical power and the generalizability of results, as observed in Nguyen et al. [41] and Valenstein-Mah et al. [40]. Short-term interventions and limited follow-ups also make it challenging to evaluate the sustainability of therapeutic effects over time. For instance, studies like Cotter et al. [42] and Vinci et al. [44] conducted interventions with only a few sessions, which might capture immediate improvements, but not long-term behavioural changes.

Future studies should address these limitations by employing group intervention formats, flexible and attractive schedules and locations for young people, extending the frequency of interventions, frequently monitoring population changes, using both objective and self-report measures, large samples, replicability in populations with different sociodemographic characteristics and contexts, randomisation of groups, separating the roles of researcher and therapist, measuring substances in the blood and conducting long-term follow-ups to evaluate intervention effects. Although the present study focused on problematic drug use, future research should also examine the application of Mindfulness, Acceptance and Commitment Therapy (ACT) and Dialectical Behaviour Therapy (DBT) for other mental health problems.

Similarly, mindfulness-based interventions require additional research to overcome the limitations identified in the analysed studies, which include reliance on qualitative self-report measures, difficulty in generalizing results, lack of study replication, inadequate control of experimental conditions, insufficient detail of procedures, lack of repeated measures, continuity in participants’ therapy involvement and the need for longitudinal and effectiveness studies. These issues may be attributed to the relatively recent emergence of third-generation therapies, resulting in fewer trials compared to other therapeutic models. However, the existing evidence is promising and suggests a need for continued research in this area [30,44,45].

It is important to note that although this review highlights a higher prevalence of mindfulness-based interventions, this does not imply an exclusion or undervaluing of other third-generation therapies, such as Acceptance and Commitment Therapy (ACT) or Dialectical Behaviour Therapy (DBT). The frequency of studies on mindfulness may stem from its growing acceptance and applicability in educational and youth mental health settings, where its effectiveness in reducing symptoms like anxiety, stress and substance use has been well-documented. This prevalence also reflects the interest in mindfulness as an accessible strategy that can be easily integrated into intervention programs without necessarily requiring highly structured therapeutic settings.

However, this review recognises the effectiveness of other third-generation interventions and emphasizes the need for future research that assesses ACT and DBT in this specific context, given their complementary potential in treating substance use and other mental health issues. By frequently presenting mindfulness in the studies, this review reflects the current trend in research without suggesting an absolute superiority over other third-generation therapeutic techniques.

## 5. Conclusions

The problem of psychoactive substance use among young adults presents implications for public health, quality of life and socioeconomic development of societies. In this context, third-generation therapies (TGTs) are a promising approach to address this problem in educational settings. The results of this scoping review highlight the efficacy of TGTs, such as ACT, DBT and mindfulness, in reducing substance use and improving emotional well-being in this population. The practical applicability of these findings is evident in clinical care and community interventions. Mental health professionals and educators can benefit from implementing TGT-based interventions to address the specific needs of young adults with psychoactive substance use problems. These therapies offer a holistic approach that addresses not only behavioural symptoms, but also the underlying emotional and psychological factors that contribute to the maintenance of addictive behaviour. Furthermore, our findings have implications beyond the clinical setting, as they could inform public health policies aimed at preventing and treating substance use in young populations. Promoting the implementation of TGT-based programs in educational settings may be an effective strategy to reduce the social and economic burden associated with future substance abuse.

## Figures and Tables

**Figure 1 behavsci-14-01192-f001:**
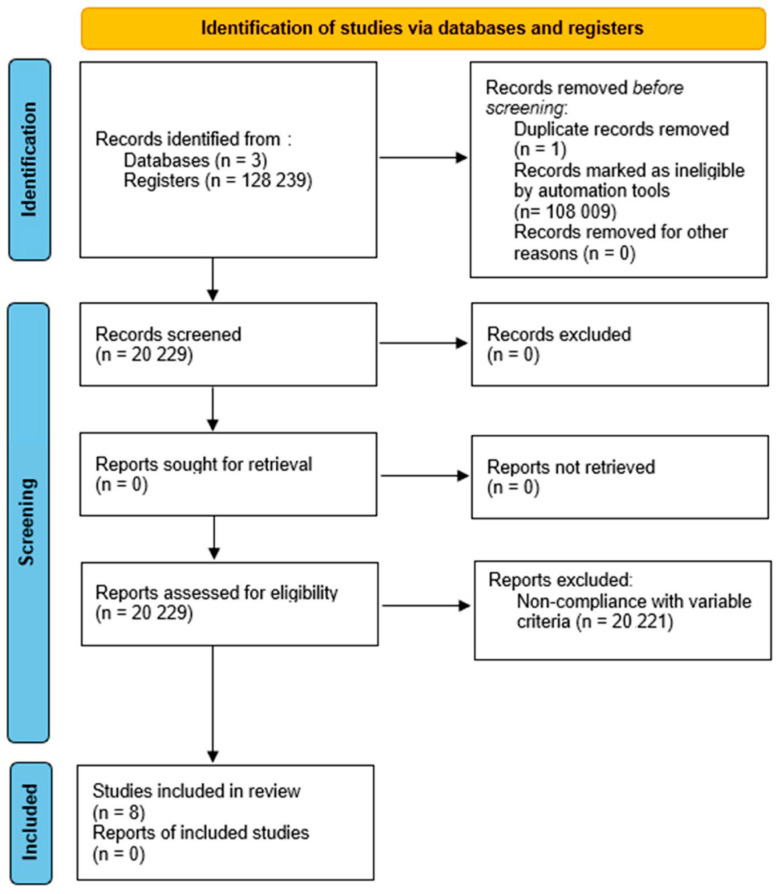
PRISMA 2020 flowchart, showing the process of search, selection, inclusion and exclusion of articles. Source: PRISMA 2020.

**Table 1 behavsci-14-01192-t001:** PIO question.

P	Problem-population	Young university students with psychoactive substance abuse
I	Intervention	Mindfulness based therapies, Acceptance and Commitment Therapy (ACT), Dialectical Behavioural Therapy (DBT)
O	Results	Protocol treatment of psychoactive substance use in young university students

**Table 2 behavsci-14-01192-t002:** DESCH and MESH descriptors.

Term	Descriptor
Third-Wave Therapies	Third-Wave Therapies, Third-Generation Therapies, Mindfulness-Based Interventions, Third Wave Psychotherapy, Mindfulness, Acceptance and Commitment Therapy, Dialectical Behavioural Therapy
Substance Use Disorders	Substance Use Disorders, Substance Dependence, Alcohol Abuse, Drug Abuse, Substance Abuse, Substance Use
Students	Students, College Students, University Students

**Table 3 behavsci-14-01192-t003:** Search algorithms.

Database	Search Algorithms
SCOPUS PUBMED	“Third-Wave Therapies” OR “Third-Generation Therapies” OR “Mindfulness-Based Interventions” OR “Third Wave Psychotherapy” OR “Mindfulness” OR “Acceptance and Commitment Therapy” OR “Dialectical Behavioural Therapy” AND “Substance Use Disorders” OR “Substance Dependence” OR “Alcohol Abuse” OR “Drug Abuse” OR “Substance Abuse” OR “Substance Use” AND “Students” OR “College Students” OR “University Students”
Web of Science	(((((((((((((((ALL = (Third-Wave Therapies)) OR ALL = (Third-Generation Therapies)) OR ALL = (Mindfulness-Based Interventions)) OR ALL = (Third Wave Psychotherapy)) OR ALL = (Mindfulness)) OR ALL = (Acceptance and Commitment Therapy)) OR ALL = (Dialectical Behavioural Therapy)) AND ALL = (Substance Use Disorders)) OR ALL = (Substance Dependence)) OR ALL = (Alcohol Abuse)) OR ALL = (Drug Abuse)) OR ALL = (Substance Abuse)) OR ALL = (Substance Use)) AND ALL = (Students)) OR ALL = (College Students)) OR ALL = (University Students)

**Table 4 behavsci-14-01192-t004:** Consolidation table *.

Database	Deleted Documents						Total Sample
	Initially Found	Type of Document	Time Period	Incomplete/Duplicated Texts	No Access	Non-Compliance with Variable Criteria	
PUBMED	55,442	52,646	2044	1	380	370	1
SCOPUS	89	13	25	0	22	25	4
Web of Science	72,708	14,854	22,024	0	16,001	19,826	3
Total	128,239	67,513	24,093	1	16,403	20,221	8

Source: own elaboration. * Database: digital search system used. Deleted documents: description of the filters used for the deleted files. Initially found: number of documents identified in the first search, before applying filters. Type of document: the types of articles selected were experimental, quasi-experimental, case study or case control, with psychological intervention based on third-generation therapies. Time period: documents from the last five years. Incomplete/duplicated texts: texts with incomplete information or repeated in the database. No access: documents identified as having access restrictions. Non-compliance with variable criteria: documents that do not meet selection criteria. Total Sample: number of data that meet the inclusion criteria.

**Table 5 behavsci-14-01192-t005:** Information from studies on third-generation psychological therapies for the treatment of substance use in young university students *.

Database	Title	Authors	Year	Method	Population	Sample	Intervention	Duration	Findings
PUBMED	Stress reduction through mindfulness meditation in Student registered nurse anesthetists	Foley and Lanzillotta-Rangeley	2021 [39]	Quasi-experimental	Student Registered Nurse Anesthetists	N = 34	Mindfulness meditation training, mediante aplicación movil.	8 weeks	Significant reductions in stress and anxiety.
SCOPUS	Feasibility pilot of a brief mindfulness intervention for college students with posttraumatic stress symptoms and problem drinking	Valenstein-Mah et al.	2019 [40]	Pilot study Diseño contorl aleatorizado	College students experience symptoms of post-traumatic stress disorder and many use alcohol problematically	N = 75	Mindfulness-based interventions MBI	4 weeks	Feasible and acceptable intervention with potential for further research. Mild to moderate feasibility and acceptability. Moderate group attendance There is a reduction in PTSD symptoms, a decrease in the amount of drinking and negative consequences of drinking. Increased awareness of the state of mindfulness.
WOS	Transforming stress program on medical students’ stress mindset and coping strategies: a quasi-experimental study	Nguyen et al.	2023 [41]	Quasi-experimental	Medical students	N = 205	Stress management program, DBT	10 weeks	Improved stress mindset and coping strategies.
WOS	A pilot mindfulness intervention to reduce heavy episodic drinking	Cotter et al.	2021 [42]	Pilot study Diseño contorl aleatorizado	College students	N = 36	Mindfulness intervention	8 weeks	Reduced heavy episodic drinking and perceived stress (not specific to medical students).
SCOPUS	Acceptance and commitment therapy and motivational interviewing in the treatment of alcohol use disorder in a college woman: a case study	Ehman et al.	2019 [43]	Case study	College women with alcohol use disorder	N = 1	Acceptance and commitment therapy (ACT) and motivational interviewing (MI)	12 sessions	Reduced alcohol consumption and improved psychological well-being.
SCOPUS	Effects of a brief mindfulness intervention on negative affect and urge to drink among college student drinkers	Vinci et al.	2014 [44]	Randomised controlled trial	College student drinkers	N = 64	Brief mindfulness intervention	4 weeks	Reduced negative affect and urge to drink.
SCOPUS	Mindful lawyering: a pilot study on mindfulness training for law students	Rosky et al.	2022 [45]	Quasi-experimental study	Law students	N = 64	Mindfulness course, “Mindful Lawyering”	13 weeks	Mindfulness training may occasion improvements in the well-being of law students.
WOS	Ultra-brief breath counting (mindfulness) training promotes recovery from stress-induced alcohol-seeking in student drinkers	Hogarth et al.	2020 [36]	Quasi-experimental	College students	N = 192	Mindfulness intervention	6-min	The breath counting versus control intervention improved subjective mood relative to baseline, attenuated the worsening of subjective mood produced by stress induction, and accelerated recovery from a stress induced increase in alcohol-seeking behaviour.

Source: own elaboration. * Database: The source or repository where the study is found (e.g., SCOPUS). Title: the name of the study, summarizing its main topic or focus. Authors: the researchers who conducted the study. Year: the year the study was published. Method: the study’s design, indicating how data were collected and analysed (e.g., quasi-experimental). Population: the group targeted or involved in the study. Sample: the number of participants included in the study, represented as “N =”. Intervention: the treatment or action applied in the study to evaluate its effects. Duration: the period during which the intervention was applied, or the study was conducted (e.g., 8 weeks). Findings: the main findings or conclusions of the study, showing the impact or effectiveness of the intervention.

**Table 6 behavsci-14-01192-t006:** Study effect sizes.

Author	Effect Size		Statistical Test	Average Change
Cotter et al., 2021 [42]	0.2 to 0.9	Low to high effect	Comparison of averages	Reduction in the number of drinks consumed on weekend nights in the intervention group.
Ehman and Gross, 2019 [43]	Does not report	Case Study	Not applicable	No quantitative changes reported.
Foley and Lanzillotta-Rangeley, 2021 [39]	Does not report	Indicates significant changes	Not reported, but significant changes are mentioned	Significant reduction in stress and anxiety among students.
Nguyen et al., 2023 [41]	0.2	Low effect	Cohen’s d	65% improvement in stress-related mindset and increase in coping strategies.
Rosky et al., 2022 [45]	0.66 to 1.32	Moderate to high effect	Cohen’s d	Improvements in stress, anxiety and depressive symptoms.
Shuai et al., 2020 [36]	0.2 to 1.0	Low to high effect	Repeated measures ANOVA, moderation analysis and significance test	Stress-induced decrease in alcohol seeking.
Vinci et al. [44]	Does not report	Indicates significant changes	Not applicable	No quantitative changes reported.
Valenstein-Mah et al., 2019 [40]	0.23 and 0.88	Low to high effect	Comparison between groups with test of significant differences	Reduction in PTSD symptoms and amount of alcohol consumption.

Source: own elaboration.

## Data Availability

The data from SCOPUS, PUBMED, and Web of Science were accessed and obtained in November 2023 and extended for analysis in March 2024.

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
