# Peer review of "Third-Generation Therapies for the Management of Psychoactive Substance Use in Young People: Scoping Review"

_behavsci, 2024, doi:10.3390/bs14121192_

Round 1
Reviewer 1 Report (Previous Reviewer 1)
Comments and Suggestions for Authors
The authors have substantially improved the manuscript and enriched it.
I have only one comment regarding the section Analysis of the articles and the flaws of the chosen articles. I find that a brief summary is missing, before the detailled description of the article-by-article analysis.
Author Response
I have only one comment regarding the section Analysis of the articles and the flaws of the chosen articles. I find that a brief summary is missing, before the detailled description of the article-by-article analysis.
Thank you for your comment. We have added a brief summary to the "Analysis of the Articles and Failures of the Selected Articles" section before the detailed description of each article. This summary provides an overview of the main findings and issues identified in the reviewed studies, facilitating a better understanding of the subsequent analysis.
Reviewer 2 Report (Previous Reviewer 2)
Comments and Suggestions for Authors
Reference: Manuscript ID: behavsci-3017871
It is highlighted in blue and italics, the parts of the text that are exposed in a literal way.
The paper titledTitle: Third generation therapies for the management of psychoactive substance use in young people: systematic review, addresses an interesting topic.
On page 1, lines 26-25, the authors show in the abstract as follows:
Objective: To determine the effect of third-generation therapies, Acceptance and Commitment Therapy (ACT), Dialectical Behavior Therapy (DBT), and Mindfulness, for the treatment of CPSA in college youth. Including articles with a 5-year window Method: A systematic, observational, and retrospective review was performed, under the PRISMA method, in Scopus, Pubmed, Web of Sciencie. Results: 8 studies were found, 6 of them based on Mindfulness, 1 on Dialectical Behavior Therapy and 1 on Acceptance and Commitment Therapy. The results of the studies are promising and emerging for the intervention of the problem. Conclusion: The interventions used show evidence in the reduction of CSPA and in the reduction of other mental health problems, such as anxiety, depression, posttraumatic stress. In addition, they allowed to increase well-being and mindfulness.
This research has no validity to response to the proposed objectives. Very simple. Let's go by parts.
1.-The authors try to carry out a systematic review of research where third-generation therapy intervention (ACT, DBT, CPSA) has been carried out in college youth
2 They carry out a search that covers 5 years
3.- In Scopus, Pubmed, Web of Sciencie
and…. Only 8 investigations meet the search requirements? In 15 minutes, I found at least 10 investigations that met these criteria, and they were done simply with Google Scholar.
With these 8 investigations, they cannot respond to the proposed research objectives. So, what authors should ask themselves is:
-Are we searching well? Even if the authors were to extract the information contained in these 8 works, given that they are not representative of any of the therapies they propose to study (they cannot be), they could not respond to the proposed objectives.
Comments on the Quality of English LanguageMinor editing of English language required
Author Response
1.-The authors try to carry out a systematic review of research where third-generation therapy intervention (ACT, DBT, CPSA) has been carried out in college youth
2 They carry out a search that covers 5 years
3.- In Scopus, Pubmed, Web of Sciencie
and…. Only 8 investigations meet the search requirements? In 15 minutes, I found at least 10 investigations that met these criteria, and they were done simply with Google Scholar.
With these 8 investigations, they cannot respond to the proposed research objectives. So, what authors should ask themselves is:
-Are we searching well? Even if the authors were to extract the information contained in these 8 works, given that they are not representative of any of the therapies they propose to study (they cannot be), they could not respond to the proposed objectives.
response to comments
hank you for your observations. We acknowledge the limitation in the number of studies identified and will repeat the search using more databases and search engines, including Google Scholar. We will review our search criteria to ensure a more comprehensive and representative collection of studies on third-generation therapies in university students.
We appreciate your feedback and will work to address these concerns.
Reviewer 3 Report (New Reviewer)
Comments and Suggestions for Authors
Third generation therapies for the management of psychoactive substance use in young people: systematic review.
Reviewer’s comments
While this research has merit and represents an important exploration of what could be extremely meaningful intervention for substance use habits among college-aged young adults, considerable revision is needed to organize the data and make the manuscript accessible to readers. Please see details comments below.
1. General comments on composition: There are substantial punctuation and grammatical errors throughout the manuscript. Common punctuation examples are excess commas and missing periods. Grammatical errors include several instances of missing conjunctions (i.e., “and” before the last word or phrase of a series), as well as others. It appears that the text has been translated by a non-English speaker. Please do a careful edit for readability, watching for awkward phrasing and inappropriate introductory phrases (i.e., misuse of “however”, “on the other hand”, and others), preferably with someone outside the research team who is a fluent in English composition.
2. General comments on organization: This study would greatly benefit from greater internal organization. Consider finding literature that shows the factors associated with substance use in college (stress, anxiety, depression, PTSD from previous life experiences) and examine the contents of the papers selected for review around that framework. As it stands, the manuscript is unfocused, and it is challenging for the reader to follow the information through to a coherent conclusion. Adhering to an internal framework will illuminate the meaningfulness of the data.
General comments on the tables: Many of the tables are not consistently internally aligned or structured.
3. Table 4: The data from this table could be consolidated and included in Figure 1. Nonetheless, the inset top row, title Deleted documents, should not include the second column, which is the total number of studies found.
4. Figure 1: This figure contains asterisks in the box for “records excluded”, but nothing to show to what the asterisks refer.
5. Table 5: The column formatting is so compressed that it is difficult to read many cells; and there is no space between vertical cells in columns, so it’s difficult to discern where one cell ends and another begins.
6. Table 6: Cell alignments in this table are not consistent. Some cells in a column are aligned to the bottom of the cell and others aligned to the middle. The text seems to not be left-justified, but set to spread across the space, which creates awkward spacing.
7. References to tables and figures within the text: It appears that Figure 1 is referenced first but does not appear in the text until after Table 4, which is not referenced specifically by name. Please make sure that the order of Tables and Figures is aligned with the order to which they are referred in the text.
Introduction
8. Page 2, lines 54-61: In the first sentence, the authors propose that substance use is increasing. Please indicate whether the percentages shared are a percentage increase, or a percentage of prevalence. Please cite sources for each sentence. Also consider providing data for more substances. If opiates are the deadliest, please share more information about them.
9. Page 2, line 59: Is death a health problem? Please consider elaborating on health problems associated with substance use and making a separate point about substance-related mortality rates among young people.
10. Page 2, lines 62-71: The paragraph sets out to describe the causes of excess substance use among college-age young adults, but the data shared is more related to the consequences of substance use in that population. Please revisit this paragraph and provide some information about the correlating factors for substance use among young people (e.g., stress, anxiety, depression, etc.).
11. Page 2, lines 72 – 77: The first sentence of this paragraph would be improved by being divided into two sentences. One sentence could focus on the range of current treatment modalities, and the second sentence would describe their limitations.
12. Page 2, lines 77- 84: Please consider giving a more detailed summary of T2G at this point in the introduction. This could be a paragraph in itself. In the current draft, you describe the effects of T3G without defining the term.
Materials and Methods
13. Page 3, line 95: “The information was then crawled…” It is unclear what this means. Please revise and describe this process with different language.
14. Page 3, lines 98 – 99: There is no need to repeat the hypothesis at this point. You could simply state that you are aligning analysis to answer the research question, which was described clearly in the Introduction.
15. Page 4, Eligibility Criteria: Please consider revising this section for greater clarity. For instance, the section beginning on line 135 would be better placed at the beginning, followed by a detailed list of inclusion and exclusion criteria.
16. Page 4, lines 131-146: The phrase “from there” is used twice in these few lines. Please review and revise to remove this duplicated phrase.
17. Page 4, lines147-148: The sentence beginning with “descriptive analysis…” is describing what will happen in the Results section and can be edited and used in that section instead of in the Methods section.
Results
18. Results, general: This section is in need of considerable revision for readability and internal organization. Several paragraphs are undeveloped and could be combined with other data points in other undeveloped paragraphs to create a more coherent narrative.
19. Page 8, lines 195 – 203: “Of the studies that met the inclusion criteria, 8 articles resulted where the therapeutic strategy in 6 studies was Mindfulness, with therapeutic strategies related to…” Please consider simplifying language in sentences like this to state your point in a straightforward manner.
20. Page 9, lines 220 – 228: Please re-organize this paragraph for greater readability. Consider grouping increased positive attributes, such as increased well-being and mindfulness, and decreases in negative attributes, such as decreased consumption of psychoactive substances, separately.
21. Page 10, line 255: “Furthermore, in the latter study…” It is not clear which is the latter study in this context. Please revise the opening of this paragraph.
Discussion
22. Discussion, general: Similar to the Results section, this portion of the manuscript needs to be revised for greater cohesion and clarity. There are many small paragraphs which could be combined and edited to make a point more clearly. Some details follow.
23. Page 11, lines 303 – 320: These two opening paragraphs could be edited and moved to the Introduction to help define the T3G.
24. General: Sometimes, the term “mindfulness” is capitalized, and sometimes it is not. If it is referring to a specific sort of instruction, please define the term and capitalize for those instances.
25. General: In the introduction, the authors refer to Third Generation Therapies as T3G, but in the Discussion, they are being called TTG. Please edit for consistency.
26. Discussion, general: If the journal formatting allows, please consider adding subsections for Limitations and Conclusions or Recommendations. Then, carefully organize the limitations and recommendations so that they can be succinctly shared in simple language in each section. In the current draft, the distinction between these sections is not clear.
Comments on the Quality of English LanguageThe language will require extensive editing.
Author Response
Dear Editor,
Please find attached our responses to the comments received on our manuscript. We have carefully considered each observation and made the necessary revisions to enhance the clarity and quality of the paper.
We sincerely appreciate the thorough review and hope our responses meet your expectations. We look forward to any further feedback you may have.
Best regards,

Round 2
Reviewer 2 Report (Previous Reviewer 2)
Comments and Suggestions for Authors
COMMENTS TO THE AUTHORS
On Monday, July 15, I received a request to review an article I had already reviewed, the following:
Manuscript ID: behavsci-3017871.
The authors sent a new version of the manuscript, now called:
Manuscript ID: behavsci-3017871-peer-review-v2
I have carefully read the new version.
I conclude as I did the first time. This manuscript, in my opinion, is categorically unpublishable.
The authors have made absolutely no substantial changes to the article.
In the first review I made, I stated the following:
This research has no validity to response to the proposed objectives. Very simple. Let's go by parts.
1.-The authors try to carry out a systematic review of research where third-generation therapy intervention (ACT, DBT, CPSA) has been carried out in college youth
2 They carry out a search that covers 5 years
3.- In Scopus, Pubmed, Web of Sciencie
and…. Only 8 investigations meet the search requirements? In 15 minutes, I found at least 10 investigations that met these criteria, and they were done simply with Google Scholar.
With these 8 investigations, they cannot respond to the proposed research objectives. So, what authors should ask themselves is:
-Are we searching well? Even if the authors were to extract the information contained in these 8 works, given that they are not representative of any of the therapies they propose to study (they cannot be), they could not respond to the proposed objectives.
I recommend restarting the investigation and carrying out a search again that provides representative results.
This paper could serve as an academic work for a subject in an academic course, but it is unsuitable for publication in a scientific journal.
In this second review, I state the following:
This research has no validity to response to the proposed objectives. Very simple. Let's go by parts.
1.-The authors try to carry out a systematic review of research where third-generation therapy intervention (ACT, DBT, CPSA) has been carried out in college youth
2 They carry out a search that covers 5 years
3.- In Scopus, Pubmed, Web of Sciencie
and…. Only 8 investigations meet the search requirements? In 15 minutes, I found at least 10 investigations that met these criteria, and they were done simply with Google Scholar.
With these 8 investigations, they cannot respond to the proposed research objectives. So, what authors should ask themselves is:
-Are we searching well? Even if the authors were to extract the information contained in these 8 works, given that they are not representative of any of the therapies they propose to study (they cannot be), they could not respond to the proposed objectives.
The authors were required to make substantial changes to the article. These changes were absolutely necessary to provide a minimum degree of validity to this research. The authors have not made any substantial changes from those that were required, I conclude that:
The research contained in this article has no validity whatsoever.
Comments on the Quality of English Language
Minor editing of English language required
Author Response
The comments are included in the attached document, which provides a point-by-point response to the various observations made.

Reviewer 3 Report (New Reviewer)
Comments and Suggestions for Authors
Third generation therapies for the management of psychoactive substance use in young people: a systematic review
1. General: The authors have done substantial revision, and some parts of the manuscript are much improved for their efforts. However, there are still several issues unaddressed throughout the study, and some of the new work needs to be refined. See details numerated below.
2. General: There are still many grammatic problems in the paper. Conjunctions and articles are still missing or incorrectly used from time to time, and there are a couple of instances with incomplete sentences. It’s not clear whether this is just an editorial omission or a problem with punctuation (i.e., using full stops [.] instead of commas [,]). Look at page 2, lines 75-76, and page 13, line415, for examples.
3. Abstract, page 1, line 23: Unless the papers in your study have a focus specifically on substance use disorder (SUD), please consider using another phrase. This phrase is used to refer to a specific diagnosis. I searched the remainder of your paper, and there are no more references to SUDs specifically within the text narrative. You could say “substance use” or “substance use behaviors” or “habitual use of illicit substances”. Alternatively, you could include more references to SUD throughout the text to indicate that you are referring to the diagnosed condition.
4. Introduction, page 2, lines 62-66: This paragraph doesn’t have much content. You could reduce it to one sentence summarizing the impact of opioids on fatal overdose (as many as 70% in some countries), and just add it to the end of the previous paragraph.
5. Introduction, page 2, line 67: It would be good to specify higher education when introducing the subject of substance use in education.
6. General: The introduction is much improved, but still needs to be polished.
7. Eligibility criteria, page 4, lines 149-151: The first point in the three-point criteria list is duplicated, and not capitalized.
8. Table 4, page 5: The current layout of Table 4 makes it look as if the “total found” documents were deleted. Please adjust the top column (“deleted documents”) so that it does not contain the “total found” column.
9. Table 5, pages 6 and 7: The title of this table is not separated from the text of the previous page (bottom of page 6).
10. Table 5: There are still formatting issues with this table. In some rows, the text is in the center of the cell, and in others it is formatted to the bottom. Please adjust this for easier reading.
11. Results, limitations of studies, pages 10-11, lines 285-311: This section has major issues. The entirety of the first paragraph is included in the second part of the second paragraph, and the entirety of the third paragraph is verbatim the same as the first part of the second paragraph.
12. Results, limitations of studies, pages 10, lines 293-305: Parts of this paragraph read more like discussion than results. Please revise to state limitations clearly and discern what analysis you want to include in the discussion.
13. Results/Discussion, general: The results spend a great deal of time on the limitations of the included studies, and not much time on the results of the studies. This is somewhat understandable, considering the small sample. However, in the discussion, the authors make broad statements, such as: “Thus, the evidence presented in this systematic review ranks mindfulness-based therapy as the most widely used and positive intervention for young people in educational settings.” It is not clear that such statements are warranted, considering the displayed results of the previous section. While all the studies showed positive effect, only 1 out of 8 presents an altogether positive effect (Rosky et al. 2022), and that one has notable risk of bias due to the researcher being a mindfulness instructor. Of the remaining studies, 3 had low to high effects, one had only low effect, and two did not report a numerical effect but showed significant change.
14. Page 12, line 359: “Mindfulness” is misspelled.
15. Discussion, general: The opening phrases of paragraphs (i.e., on the other hand, in this context, however) are often mismatched with the content that follows. Please consider whether the phrases used to begin a paragraph are necessary and whether they contribute to the meaning you are trying to convey.
16. Discussion, page 12, lines 372-388: The two paragraphs about DBT might be better combined into one paragraph.
17. Discussion, page 13, lines 405-420: Consider combining these two paragraphs into one that contains both the value and the limitations of the current study. The first paragraph right now consists of one sentence, and it’s not clear what the context is.
18. Discussion, page 13, limitations, lines 409-420: Please compare this paragraph with the similar section in the Results and see what can be combined for analysis of limitations. In this case, the limitations of the included studies are a part of the limitations of the systematic review.
19. Discussion, page 13, lines 421-429: The first sentence, beginning with “These include” seems to begin abruptly, without context. By “these”, do you mean the interventions? Other than the abrupt introduction, this paragraph is good.
20. Conclusion, general: Again, be careful that your conclusion is reflective of your results.
21. Conclusion, page, 14, lines 460+: In terms of future research, you have also stated elsewhere in the paper that greater sample size and more standardized measures for studying outcomes from TGTs are also needed to help strengthen the case for these kinds of interventions.
Author Response
Dear Reviewer,
Thank you for your valuable comments. We have carefully revised the manuscript to address your observations.
We appreciate your feedback and hope these revisions meet your expectations.
Sincerely,
Johan Acosta-Lopez

Round 3
Reviewer 2 Report (Previous Reviewer 2)
Comments and Suggestions for Authors
COMMENTS TO THE AUTHOS
Now, the authors have sent the third version of the manuscript, now called:
behavsci-3017871-peer-review-v3
The researchers do not understand the messages I sent in the previous reviews.
They cannot call a systematic review the review of 8 investigations that share search characteristics. One thing is that they have carried out the review as required by the canons in the PRISMA standard, and another thing is that the search has been effective.
Many investigations show the limitations of this type of search, and they should refer to it because they suffer it now in this research.
In the response that the reviewers give to my critic review, they write:
Regarding the selection of databases, we chose Scopus, PubMed, and Web of Science due to their international recognition, rigor, and the high quality of research they host. These databases cover a broad range of relevant research in the fields of mental health and substance use, enabling us to conduct a robust review. We acknowledge that this approach may have excluded relevant studies from other databases, which we recognize as a limitation of our study. Nevertheless, we believe that the choice of these databases is appropriate for meeting the objectives of our review. For future research, we suggest expanding the search to include other databases that may provide additional or complementary studies.
However, in the discussion section, on lines 441-471, the authors write:
Regarding the limitations of the analyzed studies, difficulties were observed in the use of objective methodologies, as some studies relied on self-report measures not compared with quantitative measures, potentially leading to symptom overestimation by participants [40, 44, 43, 45, 38, 41, 39, 42]. Additionally, in some studies, post-intervention measurements were not taken, hindering knowledge about the maintenance of improvements [43, 38, 41]. The use of mobile applications [51] and brief interventions [41, 42], coupled with small sample sizes, limits the generalizability of results [40, 44, 43, 38, 39, 42]. In general, the studies exhibit various limitations, such as a lack of detailed description of the outcome assessment process, difficulties in controlling variables, unclear criteria for determining the interval between interventions, and the presence of a single researcher acting as both evaluator and therapist.
Future studies should address these limitations by employing group intervention formats, flexible and attractive schedules and locations for young people, extending the frequency of interventions, frequently monitoring population changes, using both objective and self-report measures, large samples, randomization of groups, separating the roles of researcher and therapist, measuring substances in the blood, and conducting longterm follow-ups to evaluate intervention effects. Although the present study focused on problematic drug use, future research should also examine the application of Mindfulness, Acceptance and Commitment Therapy (ACT), and Dialectical Behavior Therapy 460 (DBT) for other mental health problems.
Similarly, mindfulness-based interventions require additional research to overcome 463 the limitations identified in the analyzed studies, which include reliance on qualitative 464 self-report measures, difficulty in generalizing results, lack of study replication, inade-465 quate control of experimental conditions, insufficient detail of procedures, lack of re-466 peated measures, continuity in participants' therapy involvement, and the need for longi-467 tudinal and effectiveness studies. These issues may be attributed to the relatively recent 468 emergence of third-generation therapies, resulting in fewer trials compared to other ther-469 apeutic models. However, the existing evidence is promising and suggests a need for con-470 tinued research in this area [52, 53, 33].
So, even though these 8 articles have successfully passed peer review, they have too many flaws, and they should not have passed the peer review process without having fixed those aspect before being published.
So, I see that this latest version of the manuscript is practically the same as the previous one. That is, the authors have not made any relevant changes. The authors insist on the same thing as the rev-2.
In my opinion, this article should only be published if the real weaknesses of the research method they have used (for the specific objective they set out) are revealed, as well as the real weaknesses of the study resulting from the 8 investigations to carry out the objectives proposed in their research. Among them, the following:
One. A systematic review was intended, but in reality, what was carried out following the standard selection criteria was only a review of a few investigations.
Two. It has been proven that the eight investigations have important methodological flaws that threaten the validity of the statistical conclusion, as well as the internal validity. Of course, with this being the case, the external validity can be questioned. How can it be said that these therapies are effective if it is impossible to separate the publication bias from the validity of the effects found when the lack of control in these investigations is manifest?
Three. Present it as a study of 8 particular investigations, and stand out the convergence between them in the results and all the weaknesses of the aforementioned investigations, making a constructive criticism about the publication bias and the bias that the weaknesses of the specific investigations imprint.
Four. They can only refer to Mindfulness therapy. Mindfulness is a third-generation therapy, but it is not representative of third-generation therapies. The authors will have to provide an explanation as to why this is the therapy that appears most frequently in this “review.” The other two therapies have an anecdotal representation.
If this is not done, my conclusion is the same as before. This article is invalid. Furthermore, it is false that a systematic review is being done and that it is about third-generation therapies. Only one review is being done, and it is only about mindfulness. The rest was left as “what was intended to be done.” It is very different from “what was finally done.” So, the title is not telling the truth. Do the authors now realize that if this research is published, and another study replicates this same research, this article could be selected from the start, and the authors would then have to remove it from the review analysis?
Author Response
Please see the attachment.

Reviewer 3 Report (New Reviewer)
Comments and Suggestions for Authors
The authors have done substantial revisions, and the manuscript is greatly improved from their efforts. They have also courteously replied to the reviewer’s comments and given thorough explanations of their process.
This is a novel study that will hopefully inspire more research into third generation therapies. I appreciate the opportunity to read this study and commend the authors on their thorough work.
I would suggest accepting this manuscript.
Author Response
We are very grateful for your valuable comments.
This manuscript is a resubmission of an earlier submission. The following is a list of the peer review reports and author responses from that submission.
Round 1
Reviewer 1 Report
Comments and Suggestions for Authors
The paper focuses in to review an interesting subject, the potential usefulness of third-generation therapies for psychoactive substance consumption.
I have some comments
Introduction:
Perhaps the first paragraph needs to be specifc for the subject, the relevance or the impact of drug consume in públic healht and the difficult to treat it.
It should be centered in the efficacy or the usefulness of psychological therapies for drug consume, the firs, second and third generation therapies. There exists more data that the authors have been introduced.
Method:
It is adecuate but it needs to specify it the review is for effectiveness or for efficacy and chek the papers obtained.
If it is a narrative reveiw the methodology will be diferent.
Results
The description of the 4 papers was adequate. There are some changes in the type of the letter (page 2, line 160)
Discussion
It will include the efficacy of psychological therapies for drug addiction, not only in the results obtained (mindfulness).
In the page 3, line 189 it is also a change in the type of the letter.
Reviewer 2 Report
Comments and Suggestions for Authors
Reference: Manuscript ID: behavsci-2763481
In my opinion, the paper presented under the title "Title: Third-Generation Therapies for Psychoactive Substance Consumption: A Systematic Review”, addresses a topic of interest.
It is highlighted in blue and italics, the parts of the text that are exposed in a literal way.
The title of the article is: Third-Generation Therapies for Psychoactive Substance Con-2 sumption: A Systematic Review
The objectives proposed by the authors are the following: Lines 62-65:
Therefore, this study aimed to explore the effectiveness of third-generation therapies—Acceptance and Commitment Therapy (ACT), Dialectical Behavior Therapy (DBT), and Mindfulness—in treating psychoactive substance use among young adults in educational contexts.
In section 2. Method, the authors write on lines 72-90:
As part of the systematic review, the following databases were consulted: ScienceDi rect, Scopus, and Web of Science. The following search equations were used: “Acceptance and commitment therapy” AND “substance use” AND “students”; “Dialectical behavior therapy” AND “substance use” AND “students”; “Mindfulness” AND “substance use” AND “students.”
The Rayyan platform was utilized in its free version, enabling the organization, filtering, and analysis of the information from the identified articles (Ouzzani, Hammady, Fedorowicz & Elmagarmid, 2016).
Research works published in English or Spanish from 2018 to 2023 were considered. The search was conducted on December 1, 2022, and the selected articles included studies with cross-sectional designs, longitudinal studies, randomized controlled trials, regression analysis, clinical studies with pretest and posttest, trials with repeated measures, and group care formats. Articles involving children or adolescents, interventions from other psychological models, evaluations of consumption, other psychological problems, scale validations, books or manuals, and interventions with populations other than young adults were excluded.
Next, the selected articles were thoroughly examined and analyzed by reviewing aspects such as title, abstract, introduction, methods, results, discussion, and other relevant information (see Figure 1).
In section 3.1. Characteristics of the studies, lines 111-127
Among the studies that met the inclusion criteria, 4 articles were identified wherein the therapeutic strategy employed was Mindfulness. This involved guided meditations conducted over 4-week sessions or through a single application, employing techniques such as loving kindness, “surfing with urgency” technique, compassionate letter, STOPP technique, and breath counting.
All four investigations focused on addressing problematic alcohol use among young adults in academic contexts. None of the studies extended their focus to a university population consuming other psychoactive substances. Variables compared alongside the Mindfulness state included post-traumatic stress, anxiety, depression, positive and negative affect, impulsive behavior, and self-improvement. Overall, the four studies consistently reported a reduction in psychoactive substance consumption, improvements in emotional states, and an increased personal well-being and mindfulness.
In terms of research methodologies, four studies employed randomized control designs (Valenstein-Mah et al., 2019; Cotter et al., 2021; Shuai et al., 2020), while one study adopted a quasi-experimental nonrandomized design (Rosky et al., 2022). All studies utilized both experimental and control groups for comparing intervention effects.
Although the topic is fascinating, the research is very deficient. For several reasons:
First and main. The four articles selected for the review have a very bad methodological quality (the data analysis in them and the control is very deficient). Therefore, the effectiveness of the intervention evaluated in each has no validity. Thus, this systematic review is not valid either because it is not helpful to achieve the proposed objective.
Second. The title should be changed. Only Mindfulness therapy is evaluated, so third-generation treatments are not being examined, but rather a single third-generation therapy, which is Mindfulness.
Third. They should question how they selected the articles. It is strange that, being the most used therapies, only four articles enter the search criteria. They must ask about the procedure and how they have done it.
Quarter. All the students are university students, so check the underlining in the previous clipping (on lines 111,127)
Comments on the Quality of English Language
Minor editing of English language required
Reviewer 3 Report
Comments and Suggestions for Authors
Thank you for the posssibility to review this paper.
The paper does not deserve publication in its current form because it systematically reviews the evidence from only four poorly retrieved papers. There is no scientific need for such a review as the impact of conclusions is close to zero. Authors should either broaden their topic or change inclusion criteria for capture more studies. I also suggest better search strategy: if the focus are young adults, what is the sense to put "AND students" in the search strategy ? Are all the young adults students? don't think so...